# Prevalence, Genetic Diversity, Antimicrobial Resistance, and Toxigenic Profile of *Vibrio vulnificus* Isolated from Aquatic Environments in Taiwan

**DOI:** 10.3390/antibiotics10050505

**Published:** 2021-04-29

**Authors:** I-Ching Lin, Bashir Hussain, Bing-Mu Hsu, Jung-Sheng Chen, Yu-Ling Hsu, Yi-Chou Chiu, Shih-Wei Huang, Jiun-Ling Wang

**Affiliations:** 1Department of Kinesiology, Health and Leisure, Chienkuo Technology University, Changhua City 500, Taiwan; licypy01@gmail.com; 2Department of Family Medicine, Asia University Hospital, Taichung City 413, Taiwan; 3Department of Earth and Environmental Sciences, National Chung Cheng University, Chiayi 621, Taiwan; bashir.aku@gmail.com; 4Department of Biomedical Sciences, National Chung Cheng University, Chiayi 621, Taiwan; 5Center for Innovative on Aging Society (CIRAS), National Chung Cheng University, Chiayi 621, Taiwan; 6Department of Medical Research, E-Da Hospital, Kaohsiung City 824, Taiwan; nicky071214@gmail.com; 7Department of Nuclear Medicine, Ditmanson Medical Foundation Chia-Yi Christian Hospital, Chiayi 600, Taiwan; cych07381@gmail.com; 8General Surgery, Surgical Department, Cheng Hsin General Hospital, Taipei 112, Taiwan; sakiochen@msn.com; 9Center for Environmental Toxin and Emerging Contaminant Research, Cheng Shiu University, Kaohsiung City 833, Taiwan; envhero@gcloud.csu.edu.tw; 10Super Micro Research and Technology Center, Cheng Shiu University, Kaohsiung City 833, Taiwan; 11Department of Internal Medicine, National Cheng Kung University Hospital, Tainan 704, Taiwan; jiunlingwang@gmail.com

**Keywords:** *Vibrio vulnificus*, virulence gene, ERIC-PCR, antibiotic susceptibility, aquatic environment

## Abstract

*Vibrio vulnificus* is a gram-negative, opportunistic human pathogen associated with life-threatening wound infections and is commonly found in warm coastal marine water environments, globally. In this study, two fishing harbors and three tributaries of the river basin were analyzed for the prevalence of *V. vulnificus* in the water bodies and shellfish that are under the pressure of external pollutions. The average detection rate of *V. vulnificus* in the river basins and fishing harbors was 8.3% and 4.2%, respectively, in all seasons. A total of nine strains of *V. vulnificus* were isolated in pure cultures from 160 samples belonging to river basins and fishing harbors to analyze the antibiotic susceptibility, virulence gene profiles, and enterobacterial repetitive intergenic consensus PCR (ERIC-PCR) fingerprinting. All isolates were susceptible to 10 tested antibiotics. The genotypic characterization revealed that 11.1% (*n* = 1/9) strain was nonvirulent, whereas 88.9% (*n* = 8/9) isolates were virulent strains, which possessed the four most prevalent toxin genes such as *vcgC* (88.9%), *16S B* (88.9%), *vvhA* (88.9%), and *manIIA* (88.9%), followed by *nanA* (77.8%), *CPS1* (66.7), and *PRXII* (44.4%). Additionally, ERIC-PCR fingerprinting grouped these nine isolates into two main clusters, among which the river basin isolates showed genetically diverse profiles, suggesting multiple sources of *V. vulnificus*. Ultimately, this study highlighted the virulent strains of *V. vulnificus* in the coastal aquatic environments of Taiwan, harboring a potential risk of infection to human health through water-borne transmission.

## 1. Introduction

*Vibrio vulnificus* is a halophilic, gram-negative, motile, curved, and rod-shaped pathogenic bacterium that belongs to the family Vibrionaceae, which naturally occurs in the coastal and estuarine environment, especially in warm temperatures (22–30 °C) and moderate saline zones (15–20‰ salinity) worldwide [1,2,3]. However, at a cold temperature (<13 °C), *V. vulnificus* turns into a viable but not culturable (VBNC) state, which makes it more resistant to various lethal environmental stress factors, as compared with the culturable cells of these bacteria [2,4,5]. Coastal and estuarine environments are aquaculture areas, and, due to the filter-feeding ability of shellfish, they have a high concentration of *V. vulnificus*. Shellfish, in turn, may become potential reservoirs for the entrance of these pathogenic bacteria into the food chain, becoming a source of wound infections in humans [6,7,8,9]. *V. vulnificus* has been frequently observed in various geographical areas in the world. Recent studies indicate that due to the global climate change, which resulted in increased surface water temperatures, enabled the global distribution and spread of *V. vulnificus* [10,11,12,13,14].

*V. vulnificus* has been classified into three biotypes, on the basis of biochemical and pathogenic characterization [15]. Among them, most biotype 1 strains belonged to the human pathogenic type; most biotype 2 strains belonged to the aquatic animal pathogenic type; moreover, biotype 3 (a hybrid of biotypes 1 and 2) can also cause human infections [10,16,17]. The consumption of raw or undercooked Mollusca and the exposure of wound to seawater are the major sources of getting varying degrees of the *V. vulnificus* lethal illnesses caused by septicemia, with high fever and chills, and wound infections resulting in tissue necrosis and severe bacteremia, with a fatality rate of >50%, especially in immuno-incompetent individuals [18,19,20,21,22]. *V. vulnificus* has a high variation rate in the strains’ virulence potential, which makes it difficult to differentiate rapidly, posing a threat to public health [23]. To overcome this problem, genotyping systems based on variation in the sequence of some loci, such as the 16S rRNA gene (types A and B correlated with clinical and environmental strains), the virulence-correlated gene (vcg), and the cytolysin⁄hemolysin gene (*vvhA*), which also serves as a primary feature to distinguish between clinical (C-) genotypes and environmental (E-) genotypes, have been developed [1,24].

*V. vulnificus* strains are generally susceptible to most veterinary and clinically used antibiotics [2,25]. However, a large proportion of resistance has been observed in various environmental niches, which are categorized into three levels: a low resistance against cefepime, kanamycin, oxytetracycline, tetracycline, nalidixic acid, and oxalinic acid; an intermediate resistance against aztreonam, streptomycin, erythromycin, vancomycin, clindamycin, ampicillin, penicillin, and gentamycin; and a complete resistance against tobramycin and cefazolin [26,27,28]. Such an increasing rate of antibiotic resistance in various environmental niches is correlated to the extensive use of antibiotics in agriculture, aquaculture, and clinical settings. These antibiotics, in turn, reach surface water bodies such as rivers and lakes through wastewater effluent, posing a potential threat to public health [10,29,30]. Consequently, not only do these water bodies play a crucial role in the dispersion of antibiotics and the development of resistant bacteria, but they also act as a potential reservoir for *V. vulnificus,* whereby these life-threatening pathogens are transmitted to the food chain and human body surface, becoming a cause of *V. vulnificus* infection [31].

The Puzih River has livestock wastewater contamination, and our previous studies show many species of bacteria with severe drug-resistant problems, such as *methicillin-resistant Staphylococcus aureus* (MRSA) and *Acinetobacter baumannii*, *Salmonella* [32,33,34,35,36]. Moreover, the downstream, estuary, and coast of the Puzih River are the biggest aquatic-culture region in Taiwan. Food chain security and health are utmost priorities for Taiwan; therefore, the surveillance of emerging pathogens and their toxin, as well as antibiotic-resistance profiling is necessary. The previous investigation of *V. vulnificus* in aquatic environments was approximately two decades ago [29]. This study aims to determine the prevalence and epidemiology of *V. vulnificus*, by analyzing the antibiotic susceptibility profile and the virulence gene pattern based on ERIC-PCR fingerprinting for conducting a detailed investigation of *V. vulnificus* in the fishing harbors and nearby river basins. Therefore, this study will also be carried out in order to provide the possible contamination sources and drug-resistant situations related to this pathogenic micro-organism.

## 2. Results

### 2.1. Detection Rate of V. vulnificus from Water and Shellfish Samples Associated with River Basin and Fishing Harbors

A total of 96 river samples, 24 seawaters, and 40 shellfish samples were collected from the Puzih river basin and two fishing harbors (DS and BD) for the detection of *V. vulnificus*. In spring and autumn, no *V. vulnificus* was detected from all the PR basins (Table 1). However, in the summer season, the detection rate of *V. vulnificus* was 25% in the estuary area (area C), 12.5% in the middle part (area B) in the vicinity to the urban area, and 0% in the upper section (area A), with a total detection rate of 12.5% in all PR basins. Area A of PR is the upstream confluence region, which is the farthest from the estuary. Similarly, in winter, the detection rate was as follows: 37.5% (area B), 25% (area C), and 0% (area A), with a total detection rate of 20.8% in the PR regions. From the water samples collected from fishing harbor areas (DS and BD), *V. vulnificus* was detected only in the autumn season (16.7%), with a total detection rate of 4.2% in all seasons. However, none of the isolates were detected from shellfish samples. The overall detection rate in the PR basin was 8.3% and 4.2% in the fishing harbor, irrespective of the seasons. During the study, nine isolates were successfully purified from eight positive sampling sites. These isolated strains were confirmed to be positive for *vvhA* (*V. vulnificus* targeting gene) by PCR detection.

### 2.2. Antimicrobial Susceptibility and Genotypic Profiling of Vibrio Vulnificus Isolates

A total of nine *V. vulnificus* isolates were subjected to an antibiotic susceptibility test against 10 antimicrobial categories. All of these isolates were susceptible to the employed categories of antibiotics, irrespective of the sampling locations, as shown in Table 2.

The status of virulent and nonvirulent strains of *V. vulnificus* in aquatic environments was checked by targeting a combination of virulence and nonvirulence genes to confirm the toxicity threat to humans, as shown in Table 3. The result revealed that 90% of the isolated strains carried a combination of various toxic genes. Among these, 88.9% (*n* = 8/9) strains showed the four most prevalent virulence genes in similar distribution (88.9%, *n* = 9), which included *vcgC* (viral correlated gene), *16S B* (encoding 30S small subunit of a prokaryotic ribosome), *vvhA* (encoding *V. vulnificus* cytolysin⁄ haemolysin protien), and *manIIA* (encoding enzyme IIA for mannitol fermentation operon), followed by three other toxin genes, which are: *nanA* (encoding N-acetylneuraminic acid lyase; 77.8%); *CPS1* (66.7), which serves as a capsular polysaccharide operon, regulating the production of polysaccharides; and *PRXII* (an arylsulfatase gene cluster; 44.4%). Additionally, 56% of the strains possessed both *16S A* (environmental type; nonvirulent) and *16S B* genes (clinical type; virulent), whereas one strain (10%) possessed both the *vcgC* (clinical type; virulent) and *vcgE* genes (environmental type; nonvirulent). In this study, the strain (10%) that belonged to the PR area B showed only non-virulence genes (environmental type), e.g., *vcgE*, *16S A*, *CPS2*, and *vvhA*.

### 2.3. Genetic Analysis of V. vulnificus Strains by ERIC-PCR Fingerprinting Combined with Genotypic Profiling

We combined ERIC-PCR typing and genetic diversity results with genotypic data to better interpret the origin/source and genetic variation of *V. vulnificus* strains. The ERIC-PCR fingerprinting successfully categorized the nine strains isolated from river basins and fishing harbors into two major clusters exhibiting a less than 40% Pearson similarity coefficient based on reference strain, sampling sites, and phylogenetic diversity (Figure 1). Cluster 1 contained 55.5% (*n* = 6/9) of the strains collected from river basins belonging to Area B (60%; *n* = 3/5) and Area C (40%; *n* = 2/5). Cluster 1 was further subdivided into two sub-clusters, assigned as clusters 1.1 and 1.2. In the sub-cluster 1.1, the nonvirulent strain primarily belonging to the river basin Area B was separated, exhibiting a less than 57% Pearson similarity in genotypic profile, compared with the virulent strains of sub-cluster 1.2. Additionally, sub-cluster 1.2, containing virulent strains, was further subdivided into A and B clusters. The virulent strains S02PR2911 and S02PR311, primarily from the river basin (Area A) and grouped into cluster A, showed 100% similarity in genotypic profile. Similarly, the strains S04PR2211 and S04PR2311, which belonged to the same sampling site (Area B) and was grouped into cluster B, exhibited almost a 90% similarity in genotypic profile. Cluster 2 contained the remaining 44.4% strains, belonging to the river basin Area B (25%; *n* = 1/4), Area C (50%; *n* = 2/4), and fishing harbors (25%); *n* = 1/4). This cluster’s strains showed a less than 62% Pearson similarity, which were further subdivided into two sub-clusters, assigned as 2.1 and 2.2, respectively. The sub-cluster 2.1 was further subdivided into two clusters, A and B. In cluster A, the strains BD-FH W211 and S04PR2711 were isolated from fishing harbors and the river basin (Area C), and were grouped into cluster A, exhibiting an 85% similarity in genotypic profile. Additionally, the strains S04PR2511 and S04PR3111, primarily from the river basin Area B and C, were grouped into profile B, showing almost a 90% similarity in genotypic profile. In contrast, the reference strain showing a less than 75% similarity was grouped separately.

## 3. Discussion

The differences in various environmental constraints such as temperature, salinity, precipitation, geographical locations, and pollution contents greatly influence the population and growth rate of *V. vulnificus* in different water bodies [13,37,38,39]. Therefore, the detection rate of *V. vulnificus* varies from 6% to 69% in different seasons, globally [40,41,42]. In this study, the detection rate of *V. vulnificus* in the river basins was higher in the winter season (20.8%) than in the summer season (12.5%), which is unique and different from what was shown in most of the previous studies [41,43]. The river site of our investigation is located in the subtropical area, where the weather temperature at the sampling time was over 33 °C in summer and approximately 22 °C in winter; thus, the winter season in Taiwan is more suitable for the survival of *V. vulnificus*, where, as previously highlighted, the optimal growth temperature ranges from 20–30 °C [9,43,44,45,46]. In this study, the overall detection rate of *V. vulnificus* during four seasons was 8.3% (including all PR regions), which is higher than what was shown in a recent study on *V. vulnificus* from a river in China [42]. Moreover, the 4.2% detection rate (in all fishing harbors) of our study aligns with the results of a previous study regarding *V. vulnificus* from five major harbors in Taiwan [47]. These findings imply that the yearly average detection rate of *V. vulnificus* (in the score year) was approximately 5% in Taiwan’s harbor environments.

The previous reports highlighted the relatively lower concentration of *V. vulnificus* in the surrounding waters, compared with waters containing oysters, shellfish, and mussels, whose filter-feeding ability when obtaining food causes a higher concentration of *V. vulnificus* in their intestines [3,8]. Interestingly, none of the *V. vulnificus* cases were isolated from the 40 shellfish samples associated with fishing harbors in this study. The highest detection rate of *V. vulnificus* from the estuary area (Area C) and urban residential area (Area B) of the river basin might be associated with the mixing of domestic waste and organic matters along with the supporting salinity parameters.

Usually, *V. vulnificus* is considered to be susceptible to most antibiotics used for human and animal treatment [10]. In this study, all of the nine isolates of *V. vulnificus* showed susceptibility against all of the tested antibiotics. The *V. vulnificus* strains’ resistance to some antibiotics has been reported; however, most of the strains were also susceptible to several antibiotics [48]. The aquatic bacteria with high drug-resistant profiles were mostly found to be Enterobacteriaceae (Especially in *E. coli*, *Salmonella*, etc.), Enterococcaceae, and Staphylococcaceae, among others [34,35,36,49,50,51]. Furthermore, most *V. vulnificus* populations with high drug-resistance were isolated from the shellfish [27,48]. A comparison of all these reports indicated the different variations of susceptibility and resistance patterns of *V. vulnificus* against a wide range of antibiotics, based on the environmental parameters or host [52,53], which need to be traced continuously.

Populations of *V. vulnificus* consist of heterogenic bacterial species that display variations in virulence and pathogenicity factors at strain level [54,55]. Not a single pathogenic gene is defined to hold the causative agent associated with *V. vulnificus* infections. Previous studies have emphasized the targeting of multiple virulence genes through PCR amplification for identification, differentiation, and typing of *V. vulnificus* at strain level [23,56]. In this study, we selected a combination of four key biomarker genes or virulence genes such as *vcgC*, *16S B*, *CPS1*, and *vvhA-1* to differentiate virulent strains of *V. vulnificus* from non-virulent strains. Previous study has also indicated that these virulence genes of *V. vulnificus* are highly specific to the virulent strains [57]. Additionally, three more virulence-associated genes, such as *nanA*, *manIIA*, and *PRXII*, were also included for risk assessment purposes, as previously suggested [20]. In this study, 90% of the isolated strains of *V. vulnificus* carried virulence genes, which were associated with both the river basins and fishing harbors. Our result is in accordance with the previous study, where the prevalence rate of virulent strains, isolated from Ariake Sea, Japan, was 90%; whereas, in Mikawa and Ise Bay, the prevalence rate was 70% [46]. Similarly, the prevalence of virulent strains in the marine environment, using tri-primer PCR based on 16S rRNA gene analysis, was 65% and for non-virulent strains, it was 35% [58]. However, in this study, the prevalence of the non-virulent strain of *V. vulnificus* was only 10%. Several studies have used *vcg* (*vcgC* and *vcgE*), *16S rRNA* (*16S A* and *16S B*), and *CPS* (*CPS1* and *CPS2*) to differentiate between the clinical (virulent) and environmental strains (nonvirulent) of *V. vulnificus*. Rosche et al. (2005) demonstrated that 90% of clinical strains carried the *vcgC* gene, whereas the detection rate of vcgE was 87% in environmental strains. In this study, the detection rate of *vcgC* was associated with 89.6% isolated strains, whereas the *vcgE* gene detection rate was 22.2%. However, one strain showed the PCR amplification of both *vcgC* and *vcgE* genes associated with the river basin. This is in accordance with the report of Warner and Oliver (2008a), where the *V. vulnificus* strain isolated from water and oysters was found to be both *vcgC*- and *vcgE*-positive. However, in a previous report, 26% of the clinical strains isolated from infected patients exhibited the *vcgE* gene [59]. Previously, the clinical strain isolated from oysters carried 76% *16S B*, and environmental strains possessed 15% *16S A* [60]. In this study, the prevalence of the *16S B* and *16S A* gene was 89.8% and 66.7%; whereas, 56% of strains were found to possess both *16S B* and *A*, simultaneously. This latter phenomenon has been reported frequently; some environmental strains of *V. vulnificus*, isolated from aquatic environments and oyster, possessed both *16S B* and *A* simultaneously [61], probably in order to meet the survival needs of *V. vulnificus* under different environmental conditions [57]. Based on the *CPS* allele 1 and 2 differentiation between virulent (clinical) and non-virulent strains, only one strain showed *CPS2* (*n* = 1/9; 11.1%), 66.7% (*n* = 6/9) possessed *CPS1*, and 22% (*n* = 2/9) of the strains did not show any of these two alleles. Additionally, none of the strains possessed both alleles simultaneously. However, the absence of both alleles has been reported in *V. vulnificus* strains isolated from different environments [60,62]. It has been clearly demonstrated that *CPS* plays an important role in virulence associated with clinical strains, whereas in environmental strain, it is mostly involved in survival mechanisms [63]. Based on the comparison of *V. vulnificus* genotyping in this study’s virulence factors, the *vcgC, 16S B, vvhA, manIIA*, and *GPS 2* genotypes were more appropriate for distinguishing virulent and non-virulent strains. This finding suggested that most *V. vulnificus* wild types are also a human health concern.

Previous studies have demonstrated the higher genetic diversity among the isolates of *V. vulnificus* from different environmental niches, using the ERIC-PCR method, which is more accurate as compared with REP-PCR [3]. In this study, we combined the toxigenic profiles of isolated strains of *V. vulnificus* with ERIC-PCR, which grouped these isolates into two main clusters based on their respective sampling sites, reference strain, and genotypic profiles. This is consistent with the result of the previous study, where the *V. vulnificus* strains isolated from the Baltic Sea region were classified into two main clusters using multi-locus sequence typing (MLST) [64]. The strains of *V. vulnificus* isolated from the river basin showed diverse heterogeneity in genotypic profiles, indicating unique sources of this pathogenic strain and implying an uneven distribution of genetic differences across each sampling site. The genetically divergent strains may also be associated with geographical distribution, invertebrate host preferences, environmental stress, and changing estuarine conditions [63]. However, in this study, due to the limited number of isolates from fishing harbors, the exact sources could not be traced, which warrants extended and continuous epidemiological surveillance. The result of ERIC-PCR typing in combination with the genotypic profile could be significantly useful in distinguishing the *V. vulnificus* strains from source tracking and the differences between their distinct environments.

## 4. Materials and Methods

### 4.1. Sampling Information

In this study, three sampling sites were investigated around the neighboring Puzih River (PR), Dongshi fishing harbor (DS), and Budai fishing harbor (BD). The PR region was divided into three sections according to the distance to the estuary, as shown in Figure 2. The upper section (area A) converged the upstream of the river. In contrast, the middle part (area B) was the urban residential area. Area C was an estuary, adjacent to DS and BD fishing harbors. From January 2016 to February 2017, 96 water samples were collected from the PR area and 24 samples from DS and BD in different seasons.

### 4.2. Pre-Treatment of Water Samples

About 300 mL of water was filtered through a 47 mm sterilized filter membrane with a pore size of 0.45 um (66191 GN-6, Pall Corporation, city, state, USA) to get the high concentration of bacteria. This concentrated membrane was eluted in 25 mL phosphate-buffered saline (PBS) and centrifuged at 2600× *g* for 30 min. In the case of shellfish samples, 10–20 g of shellfish meat was suspended in a 50 mL centrifuge tube along with 1S PBS, with a total volume of 40 mL. Finally, homogenization was carried out twice using an ultra-turrax tube drive (UTTD, IKA, Staufen, Germany) at a maximum rotating speed of 30 s.

### 4.3. Enrichment, Cultivation, and Molecular Profiling of V. vulnificus

The pre-treated water sample and 1 mL shellfish homogenized liquid sample was transferred into 10 mL Alkaline Peptone Water (APW; Taiwan Prepared Media, Taipei, Taiwan), followed by incubation at 30 °C, for 24 h, for enrichment. The next day, a loopful of broth from the pre-enrichment step was streaked on CHROMagar Vibrio (CV; Taiwan Prepared Media, Taipei, Taiwan) plates and Thiosulfate Citrate Bile Salt Sucrose (TCBS; Taiwan Prepared Media, Taipei, Taiwan) agar plate, followed by incubating at 37 °C, for 24 h. The next day, a single colony from agar plate was picked with the help of a toothpick and inoculated into an APW tube and then incubated at 30 °C, for 24 h. After incubation, 300 µL culture containing candidate isolates were mixed with 700 µL 33% glycerol into a 1.5 mL centrifuge tube and stored at −20 °C. The reference strain source in this study was *V. vulnificus* ATCC27562, which served as a control for subsequent experimental analysis. The DNA extraction from overnight culture was centrifuged at 10,000× *g* for 5 min and removed from the supernatant. DNA from concentrated pellet was extracted for molecular analysis using ZP02006 MagPurix automatic DNA extraction system (Zinexts Life Science Corp, New Taipei, Taiwan) provided with the bacterial DNA extraction kit, following the procedure of the manufacturer’s instructions, with the final elution in 100 µL. This eluent was then used for PCR experiments with a total reaction volume of 25 µL by adding the appropriate concentration of template, primers, and master mix, respectively, as shown in Table 1. PCR amplification was performed using Life ECO Thermal Cycler (Bioer Technology Co., Ltd., Hangzhou, China) according to the conditions of the PCR programs, as shown in Table 4. Finally, DNA quality was assessed visually by running gel electrophoresis with 1.5% 1 × TAE Buffer at 110 V, for 30 min, and bands were visualized under UV. The positive samples were identified by targeting the *vvhA* gene in PCR, which is a common gene marker for the identification of *V. vulnificus* species. Subsequently, the detected five major virulence-associated genes, including *V. vulnificus* cytolysin⁄hemolysin gene (*vvhA*), viral correlated gene (*vcg*), capsular polysaccharide operon (*CPS*), pathogenicity region XII (*PRXII*), Sial acid catastrophe region (*nanA*), and enzyme IIA of mannitol fermentation operon (*manIIA*) were used for phylogenetic analysis and strain typing, by means of ERIC-PCR fingerprinting and with the aid of commercial software BioNumerics (Applied Maths NV, Inc., Sint-Martens-Latem, Belgium). Finally, the cluster analysis was performed using curve-based Pearson correlation, and the resulting dendrogram was generated based on the unweighted pair group method with arithmetic mean (UPGMA).

### 4.4. Antibiotic Susceptibility Testing and Multidrug Resistance Profiling of V. vulnificus Isolates

The antibiotic susceptibility testing of all isolates of *V. vulnificus* was carried out using the disc diffusion method, following the guidelines of Clinical and Laboratory Standards Institute instructions (CLSI, 2010). The isolated strains of *V. vulnificus* were cultured in 5 mL Cation Adjusted Mueller Hinton Broth (CAMHB; Dr. Plate Biotech Company, Taipei, Taiwan), for 16–25 h, at 35 °C. The bacterial suspension was adjusted to 0.5 McFarland and evenly streaked on Mueller Hinton Agar (MHA). Subsequently, we aseptically placed the selected antibiotic paper disc on evenly streaked MHA and incubated at 34 °C, for 16–20 h. Finally, the antibiotic susceptibility of different strains was observed by measuring the size of the zone of inhibition (ZOI). The criteria for MDR was defined as non–susceptibility to at least one agent in three or more antimicrobial categories [67].

## 5. Conclusions

The detection rate of *V. vulnificus* strains was higher in downstream of the river basins adjacent to the residential area followed by the estuary area and fishing harbors. Additionally, none of the isolates were purified from the upper section of the river basin and the shellfish samples of these fishing harbors. Furthermore, we could isolate only one strain from the water sample of fishing harbors. These data suggest that the virulent strains of *V. vulnificus* might be enriched and spread from the surrounding urban, residential areas, most probably through domestic waste discharge mixing into the river basin. Consequently, following the water flow, this might also lead to the contamination of downflow areas, including the estuary area of the river basin and the fishing harbors. Notably, 88.9% isolated strain of *V. vulnificus* exhibited multiple virulence factors. The comparison of *V. vulnificus* genotyping based on this study’s virulence factors, the *vcgC*, *16S B*, *vvhA*, *manIIA*, and *GPS 2* genotype were more appropriate for distinguishing virulent and non-virulent strains. Fortunately, all *V. vulnificus* isolates in this study were susceptible to all tested antibiotics. The ERIC-PCR fingerprinting revealed heterogeneity among the isolated strains even at a single sampling site, the river basin. This defined the broad distribution of genetic differences across sampling sites and respective isolated strains, indicating multiple sources of these strains. Even at individual strain-level, these virulent strains of *V. vulnificus* exhibit multiple toxigenic profiles in the aquatic environments, which is a significant threat to human health. Therefore, a continuous monitoring of these coastal aquatic environments and their adjacent waste discharge areas is warranted in order to timely prevent the further spread of *V. vulnificus*.

## Figures and Tables

**Figure 1 antibiotics-10-00505-f001:**
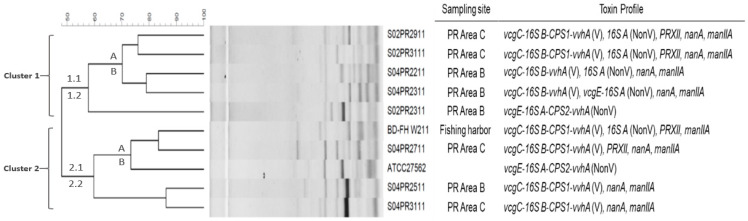
Genetic diversity of *Vibrio vulnificus* isolates by ERIC-PCR, combined with toxigenic profiling.

**Figure 2 antibiotics-10-00505-f002:**
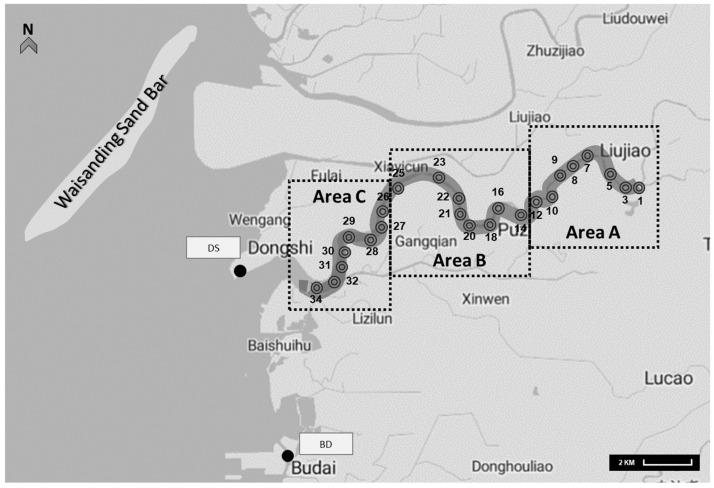
The illustration of the sampling sites and the Puzih River tributaries. The Puzih River is located in a subtropical area under the influence of high population density and significant numbers of aquaculture. The sampling sites of surface water in PR (◎) and fishing harbors (●) are marked accordingly.

**Table 1 antibiotics-10-00505-t001:** Detection rates of *Vibrio vulnificus* in aquatic water bodies and shellfish in different seasons.

Seasons	DS & BD Fishing Harbor (HW) (Shellfish)	DS & BD Fishing Harbor (HW) (Water)	Area A of PR (Water)	Area B of PR (Water)	Area C of PR (Water)	Sum of PR (Water)
Spring	0/10 (0%)	0/6 (0%)	0/8 (0%)	0/8 (0%)	0/8 (0%)	0/24 (0%)
Summer	0/10 (0%)	0/6 (0%)	0/8 (0%)	1/8 (12.5%)	2/8 (25%)	3/24 (12.5%)
Autumn	0/10 (0%)	1/6 (16.7%)	0/8 (0%)	0/8 (0%)	0/8 (0%)	0/24 (0%)
Winter	0/10 (0%)	0/6 (0%)	0/8 (0%)	3/8 (37.5%)	2/8 (25%)	5/24 (20.8%)
Total	0/40 (0%)	1/24 (4.2%)	0/32 (0%)	4/32 (12.5%)	4/32 (12.5%)	8/96 (8.3%)

**Table 2 antibiotics-10-00505-t002:** Prevalence of antibacterial susceptibility in *Vibrio vulnificus* isolates.

Strains	Zone of Inhibition (mm)									
	Ampicillin	Amoxycillin-Clavulanic Acid	Ampicillin-Sulbactam	Cefepime	Chloramphenicol	Ciprofloxacin	Gentamicin	Imipenem	Tetracycline	Trimethoprim-Sulfamethoxazole
S02PR2311	29	25	25	23	25	19	20	23	22	25
S02PR2911	21	22	22	23	25	26	20	23	23	23
S02PR3111	23	26	23	28	32	32	24	34	33	33
S04PR2211	35	28	27	31	36	34	23	38	32	32
S04PR2311	26	30	29	32	31	32	22	29	34	36
S04PR2511	28	26	20	26	25	26	20	26	19	23
S04PR2711	19	21	23	22	33	28	22	19	22	25
BD-FH W211	23	22	23	31	34	37	23	27	26	28
S04PR3111	19	19	19	25	26	31	25	22	28	24
Resistant	≤13	≤13	≤11	≤14	≤12	≤15	≤12	≤13	≤14	≤10
Intermediate	14–16	14–17	12–14	15–17	13–17	16–20	13–14	14–15	15–18	11–15
Susceptible	≥17	≥18	≥15	≥18	≥18	≥21	≥15	≥16	≥19	≥16

**Table 3 antibiotics-10-00505-t003:** Percentages of *Vibrio vulnificus* strains carrying various virulent and nonvirulent genes.

Strain	Sampling Type	Virulent Type				Nonvirulent Type				*PRXII*	*nanA*	*manIIA*
		*vcgC*	*16S B*	*CPS1*	*vvhA*	*vcgE*	*16S A*	*CPS2*	*vvhA*			
S02PR2911	PR Area C	+	+	+	+		+			+	+	+
S02PR3111	PR Area C	+	+	+	+		+			+	+	+
S04PR2211	PR Area B	+	+		+		+				+	+
S04PR2311	PR Area B	+	+		+	+	+				+	+
S02PR2311	PR Area B					+	+	+	+			
BD-FH W211	Fishing harbor	+	+	+	+		+			+		+
S04PR2711	PR Area C	+	+	+	+					+	+	+
S04PR2511	PR Area B	+	+	+	+						+	+
S04PR3111	PR Area C	+	+	+	+						+	+
Toal		8/9 (88.9%)	8/9 (88.9%)	6/9 (66.7%)	8/9 (88.9%)	2/9 (22.2%)	6/9 (66.7%)	1/9 (11.1%)	1/9 (11.1%)	4/9 (44.4%)	7/9 (77.8%)	8/9 (88.9%)

“+” indicated the presence of genes in the table.

**Table 4 antibiotics-10-00505-t004:** PCR primers and conditions for targeting toxin gene profiles, and the identification and differentiation of *V. vulnificus*.

Target Gene	Size	Sequence (5′ to 3′)	Reaction Materials Final Volume: 25 μL	PCR Condition	Reference
*vvhA*	505	FDAvvhA-F: 5′-CCGCGGTACAGGTTGGCGCA-3′ FDAvvhA-R: 5′-CGCCACCCACTTTCGGGCC-3′	DNA: 100–300 ng Primer: 300 nM Master mix: 5 μL	Pre-denaturation: 94 °C 3 min Denaturation: 94 °C 60 s Annealing: 60 °C 60 s Extension: 72 °C 60 s D.A.E. Cycles: 30 cycles Final extension: 72 °C 10 min	[65,66]
ERIC	-	ERIC1R: 5′-ATGTAAGCTCCTGGGGATTCAC-3′ ERIC2: 5′-AAGTAAGTGACTGGGGTGAGCG-3′	DNA: 100–300 ng Primer: 5000 nM Master mix: 5 μL	Pre-denaturation: 95 °C 7 min Denaturation: 92 °C 45 s Annealing: 54 °C 60 s Extension: 70 °C 10 min D.A.E. Cycles: 35 cycles Final extension: 72 °C 20 min	[3]
Virulent type *vcgC* *vcgC* *16S B* *CPS1*	99 278 839 342	vcgC-F: 5′-AGCTGCCGATAGCGATCT-3′ vcgC-R: 5′-TGAGCTAACGCGAGTAGTGAG-3′ vcg-P1: 5′-AGCTGCCGATAGCGATCT-3′ vcg-P3: 5′-CGCTTAGGATGATCGGTG-3′ 16S B-F1: 5′-GCCTACGGGCCAAAGAGG-3′ 16S B-R1: 5′-CCTGCGTCTCCGCTGGCT-3′ CPS1HP-1F: 5′-TTTGGGATTTGAAAGGCTTG-3′ CPS1HP-1R: 5′-GTGCCTTTGCGAATTTTGAT-3′	DNA: 100–300 ng Primer: 300 nM vcgC-FR, 200 nM vcg-P13, 200 nM 16S B-FR, 700 nM CPS1HP-FR Master mix: 5 μL	Pre-denaturation: 95 °C 5 min Denaturation: 94 °C 60 s Annealing: 56 °C 60 s Extension: 72 °C 60 s D.A.E. Cycles: 30 cycles Final extension: 72 °C 7 min	[21]
Nnvoirulent type *vcgE* *16S A* *CPS2*	278 839 152	vcg-P2: 5′-CTCAATTGACAATGATCT-3′ vcg-P3: 5′-CGCTTAGGATGATCGGTG-3′ 16S A-F2: 5′-AGCTTCGGCTCAAAGAGG-3′ 16S A-R2: 5′-CCAGCGTCTCCGCTAGAT-3′ CPS2HP-2F: 5′-TTCCATCAAACATCGCAGAA-3′ CPS2HP-2R: 5′-CTTTTGTCCGGCTTCTATCG-3′	DNA: 100–300 ng Primer: 300 nM vcg-P23, 300 nM 16S A-FR, 200 nM CPS2HP-FR Master mix: 5 μL	Pre-denaturation: 95 °C 5 min Denaturation: 94 °C 60 s Annealing: 50 °C 60 s Extension: 72 °C 60 s D.A.E. Cycles: 30 cycles Final extension: 72 °C 7 min	[57]
Virulent type *vvhA-1*	814	vvhA-1F: 5′-AGATTAAGTGTGTGTTGCACACAAGCGGTG-3′ vvhA-1R: 5′-ACCGAAAACAGCGCTGAAGGAAGAACGGTA-3′	DNA: 100–300 ng Primer: 400 nM Master mix: 5 μL	Pre-denaturation: 95 °C 2 min Denaturation: 95 °C 30 s Annealing: 57 °C 30 s Extension: 72 °C 90 s D.A.E. Cycles: 30 cycles Final extension: 72 °C 3 min	[22]
Nnvoirulent type *vvhA-2*	814	vvhA-2F: 5′-AAATTAAGTGCGTGCTACACACAAGTGGTG-3′ vvhA-2R: 5′-ACTGAGAAGAGTGCTGAAGGGATTACCGTA-3′	DNA: 100–300 ng Primer: 400 nM Master mix: 5 μL	Pre-denaturation: 95 °C 2 min Denaturation: 95 °C 30 s Annealing: 57 °C 30 s Extension: 72 °C 90 s D.A.E. Cycles: 30 cycles Final extension: 72 °C 3 min	[22]
*PRXII*, *nanA*, *manIIA*	2257 1299 243	VVA1612F: 5′-ACCCTGATCGTTGGCTACTC-3′ VVA1613R: 5′-GGAGCGGTGTGATGGTGTTG-3′ rpiR-F: 5′-TACGCAAGCCCAGCGGCATG-3′ nanA-2R: 5′-TTGCCACTTCCGCGATCGGG-3′ ManIIA-F: 5′-GATGTTGGTGAACAACTTCTCTGC-3′ ManIIA-R: 5′-TCTGAAGCCTGTTGGATGCC-3′	DNA: 100–300 ng Primer: 800 nM VVA-FR, 200 nM nanA-FR, 200 nM ManIIA-FR Master mix: 5 μL	Pre-denaturation: 94 °C 4 min Denaturation: 94 °C 30 s Annealing: 63 °C 30 s Extension: 72 °C 2.5 min D.A.E. Cycles: 30 cycles Final extension: 72 °C 10 min	[1]

## Data Availability

The data presented in this study are available on request from the corresponding author.

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
