# Peer review of "Prevalence, Genetic Diversity, Antimicrobial Resistance, and Toxigenic Profile of Vibrio vulnificus Isolated from Aquatic Environments in Taiwan"

_antibiotics, 2021, doi:10.3390/antibiotics10050505_

Round 1

Reviewer 1 Report

I inserted my comments in the attached file.

Author Response

Response to Reviewer 1 Comments

Point 1: Abstract section; A total of 9 isolates of V. vulnificus were purified in pure cultures from 160 samples belonging to river basins and fishing harbors to analyze the antibiotic susceptibility, virulence gene profiles, and enterobacterial repetitive intergenic consensus PCR (ERIC-PCR) fingerprinting. The word "isolates" needs to replace with "samples" and "purified" with "isolates"

Response 1: Thanks for reviewer’s comment, we have changed as suggested.

Point 2: The genotypic characterization revealed that 11.1% (n=1/9) strain was nonvirulent, whereas, 88.9% (n=8/9) isolates were virulent strains which possessed the four most prev-alent toxin genes such as vcgC (88.9%), 16S B (88.9%), vvhA (88.9%), and manIIA (88.9%), followed by nanA (77.8%), CPS1 (66.7), and PRXII (44.4%). The gene names must be italic.

Response 2: Thanks for reviewer’s comment, we have changed the gene names throughout the manuscript as suggested

Point 3: result section; Among these, 88.9% 146 (n=8/9) strains showed the four most prevalent virulence genes in similar distribution 147 (88.9%, n = 9), which included vcgC, 16S B, vvhA, and manIIA, followed by other three 148 toxin genes, nanA (77.8%), CPS1 (66.7), and PRXII (44.4%). "I suggest to describe the characteristic of these virulence genes, for which proteins codyfing?

Response 3: Thanks for reviewer’s comment, we appreciate the suggestion. We have added the information associated to all functional gene as suggested in the result section, furthermore we also have described in introduction and discussion sections.

Point 4: The table is not completely visible.

Response 4: Thanks for reviewer’s comment, we have made both tables legible.

Point 5: why genes PRXII, nanA, and manII are separated from other genes?

Response 5: Thanks for reviewer’s comment, these three genes have shown separately because they were randomly presented in both of virulent and non-virulent strains of V. vulnificus.

Point 6: The figure 1 must be larger and with a higher resolution.

Response 6: Thanks for reviewer’s comment, we have improved the quality of Figure 1.

Point 7: This section may be divided by subheadings. It should provide a concise and precise

description of the experimental results, their interpretation, as well as the experimental

conclusions that can be drawn. As per reviewer this sentence must need to remove.

Response 7: Thanks for reviewer’s comment, we have removed this sentence from the main manuscript. We are apologized that this sentence was mistakenly put inside the manuscript

Point 8: Moderate English changes required

Response 8: Thanks for reviewer’s comment, we have decided to get the MDPI English editing service after the completion of revision or when reviewer ask for editing at any time.

Reviewer 2 Report

Dear Authors,

I think, this is an quite good manuscript. The results of this study provide new knowledge for scientists and for practitioners. These results bring some knowledge of the given topic. Major corrections should be made and it should be re-reviewed:

  1. Introduction part is very poor. It should be added why this topic is important. What does the presented research bring to science? What is different from previously published?
  2. It would be good to cite publications describing drug-resistant bacteria isolated from environment in the discussion. These strains are a dangerous reservoir of drug resistance. Publications in this area that can be referred to :

Phenotypic and molecular assessment of drug resistance profile and genetic diversity of waterborne Escherichia coli. Water, Air Soil Pollution, 227:146. DOI: 10.1007/s11270-016-2833-z.

Antimicrobial resistance and the presence of extended-spectrum-beta-lactamase genes in Escherichia coli isolated from the environment of horse riding centers. Environmental Science and Pollution Research, 25:21789-21800. DOI: 10.1007/s11356-018-2274-x.

  1. Tables 2 and 3 look cut and I'm not sure if they contain all the data, please correct it or edit it legibly.
  2. Figure 1 is catastrophically poor quality, please change to better
  3. Extensive editing of English language and style required. - Please indicate the amendments in the text

Author Response

Response to Reviewer 2 Comments

Point 1: Introduction part is very poor. It should be added why this topic is important. What does the presented research bring to science? What is different from previously published?

Response 1: Thanks for the reviewer’s comment; we have improved the introduction section according to your suggestion and recommendation

Point 2: It would be good to cite publications describing drug-resistant bacteria isolated from environment in the discussion. These strains are a dangerous reservoir of drug resistance.

Response 2: Thanks for the reviewer’s comment; we have cited more literatures describing drug-resistant bacteria in surface water environments.

Point 3: Tables 2 and 3 look cut and I'm not sure if they contain all the data, please correct it or edit it legibly.

Response 3: Thanks for the reviewer’s comment; we have made both tables legible.

Point 4: Figure 1 is catastrophically poor quality, please change to better

Response 4: Thanks for the reviewer’s comment; we have improved the quality of figure 1.

Point 5: Extensive editing of English language and style required. - Please indicate the amendments in the text

Response 5: Thanks for the reviewer’s comment; we have decided to get MDPI English editing service after the completion of revision or when reviewers ask for editing.

Round 2

Reviewer 2 Report

Accept in present form